# Typologies of childhood maltreatment and associations with internalizing symptoms among university students in Singapore: A latent class analysis

**Jungup Lee**[1,2*], **Yogeswari Munisamy**[1], **Tan Ai**[3], **Sungwon Yoon**[4,5], **Jinyung Kim**[6], **Alicia Pon**[1]

**1** Department of Social Work, National University of Singapore, Singapore, Singapore, **2** Social Service Research Centre, National University of Singapore, Singapore, Singapore, **3** Child Protective Service, Ministry of Social and Family Development, Singapore, Singapore, **4** Health Services and Systems Research, Duke-NUS Medical School, Singapore, Singapore, **5** Centre for Population Health Research and Implementation, SingHealth Regional Health System, Singapore, Singapore, **6** School of Social Work, University of Maryland, Baltimore, Maryland, United States of America

* swklj@nus.edu.sg

## Abstract

This study used Latent Class Analysis to identify typologies of childhood maltreatment (CM) and the associations of CM with five internalizing symptoms. A sample of 1,042 university students in Singapore answered online self-report questionnaires, inclusive of Childhood Trauma Questionnaire, a modified version of the 14-item Center for Epidemiologic Studies Depression Scale derived from the CES-D, Beck Anxiety Inventory, Post-traumatic Stress Disorder Checklist, Eating Disorder Examination-Questionnaire (version 6.0), and Suicidal Ideation Attributes Scale. These measures respectively assessed CM and current internalizing symptoms, namely, depressive symptoms, anxiety symptoms, PTSD, eating disorder, and suicidal ideation. The most common type of CM was childhood emotional neglect (74.6%), followed by childhood emotional abuse (61%). Men were more likely to experience childhood physical abuse compared to women; contrarily, women were two times more likely to report childhood emotional abuse compared to men. The findings of Latent Class Analysis revealed four distinct latent classes of CM: *Low CM*, *high/multiple CM*, *moderate to high abuse/victimization*, and *moderate to high neglect*. Students in the latter three CM classes were more likely than those in the *Low CM* class to report the internalizing symptoms. These findings indicate the importance of protecting children from CM and cushioning the adverse effects of CM on victims by providing timely intervention, both of which would be best achieved with the education of professionals, caregivers and the public alike, and improvements to current programs and practices.

## Introduction

Childhood maltreatment (CM), which often include exposure to abuse and neglect and being victims of bullying in childhood, are a pervasive societal concern all over the world [1].

**Data availability statement:** All relevant data are within the paper and its Supporting Information files.

**Funding:** IRB-approval (LS-18-201) was obtained from the Human Subjects Committee at National University of Singapore. This study was supported by Start-Up Grant [Grant No. R134-000-098-133] from the National University of Singapore. The funders had no role in study design, data collection and analysis, decision to publish, or preparation of the manuscript.

**Competing interests:** The authors have declared that no competing interests exist.

According to the National Survey of Children's Exposure to Violence (NatSCFV), about 15.2% of children (0-17 years) in the U.S. reported child maltreatment in 2014 [2]. In Singapore, the prevalence of child protection cases by physical abuse, sexual abuse, and neglect has exponentially increased in the past decade. The cases of physical abuse, sexual abuse, and neglect escalated from 188 cases, 58 cases, and 144 cases in 2010 to 788 cases, 443 cases, and 910 cases in 2021, respectively [3].

A substantial body of literature indicates that CM consistently predict a variety of internalizing and externalizing symptoms in later life [4–7]. Internalizing symptoms refer to inner-directed and generated psychological distress that an individual faces such as depression, anxiety, post-traumatic symptoms (PTSD) and suicidal ideations, while externalizing symptoms are characterized as externally-focused behavioral symptoms involved in the surrounding environment including aggression, deviant and criminal behavior, conduct problems, and poor social functioning [8,9]. Specifically, these maltreatment experiences as a child are significantly associated with an increased risk of young adults' internalizing symptoms [10–13]. For instance, Wu et al [14] indicated that higher exposure to childhood trauma escalated the risk of PTSD and psychological distress.

Despite mounting evidence about the association between CM and internalizing symptoms, there is a paucity of empirical knowledge on how different patterns of CM influence problematic internalizing symptoms in young adulthood. For example, only a few studies determined the relationship between the typologies of adverse childhood experiences (ACEs) and internalizing problems/disorders (see Lew & Xian[4] for U.S. children). Assessing typologies of CM and its associations with internalizing symptoms has become an incumbent task for researchers where the knowledge would guide the direction of future studies, such as exploring preventive measures and effective intervention for high-risk groups. For professionals like clinicians, teachers, and caregivers of individuals who have experienced CM, understanding the typologies can guide them in directing resources toward those who need the most support, while also targeting the specific internalizing symptoms they are most likely to develop. Using a latent class analysis (LCA), the primary aims of this study are to identify the main different patterns of traumatic events that occur in childhood and how each type of CM is associated with internal symptoms among young adults, with the hope that the findings can promote efficient services for children and youth who have suffered from CM.

## Typologies of childhood maltreatment

Adverse childhood experiences (ACEs) encompass direct traumatic experiences during childhood: physical abuse, sexual abuse, emotional abuse, physical neglect, emotional neglect, as well as experiences of indirect exposure to household dysfunction during childhood, such as household substance abuse, mental illness, witnessing violence towards one's mother, history of incarceration and parental separation or divorce [15,16]. This study focuses exclusively on assessing direct maltreatment experiences in childhood, which can be classified into childhood abuse (i.e., physical, sexual, and emotional abuse), childhood neglect (i.e., physical and emotional neglect), and childhood bullying victimization (i.e., traditional bullying with physical, verbal, and relational incidents and cyberbullying via digital technologies).

Prior evidence has revealed various types of maltreatment experiences in childhood. As CM is an intricate social phenomenon, LCA can be beneficial to classify distinct patterns of CM [17]. As a person-centered approach, LCA is used to extract latent classes of co-occurring CM [18]. Few studies have used LCA to establish typologies of maltreatment experiences in childhood [19–21]. Armour et al [19] identified four typologies of CM (i.e., non-abused group, psychologically maltreated group, sexually abused group, and multiple abused group) from their sample of 2,980 Danish residents. Ballard et al [20] studied a sample

of 1,815 young adults and their behavioral outcomes in young adulthood. Using nine traumatic indicators before the age of 13, three latent classes of childhood trauma exposure were established: low levels of childhood trauma, experiences of sexual assault, and experiences of violence exposure. Keane et al [22] used a sample of 1,682 Australians to identify childhood trauma experience (CTE) and revealed six distinct classes, including multiple CTE, distal CTE, proximal CTE, high violence CTE, indirect CTE, and low CTE.

## Childhood maltreatment and internalizing symptoms in young adulthood

CM is closely linked with an escalated risk of internalizing problems in adulthood for the local community, as well as for the global society [1,11,23,24]. Many empirical investigations have demonstrated that CM contributes to a higher risk for a wide range of internalizing problems, such as depression, anxiety, PTSD, eating disorder, and suicidal activity [4,11,25–27]. Björkenstam et al [10] studied 478,141 Swedish individuals regarding the impact of childhood adversities (CAs) on mental health in young adulthood and indicated that all CAs predicted depression. Similarly, a study of 6,126 Singaporeans by Subramaniam et al [28] revealed that ACEs increased the risk of major depressive disorder, anxiety disorder, and suicidality. A meta-analysis of 35 studies conducted by Li et al [29] also found that child sexual abuse had a significant relationship with depression in female samples and clinical samples in Singapore.

Several studies have scrutinized the association between CM and internalizing symptoms in college student samples and revealed that CM was positively associated with internalizing symptoms [30–32]. For instance, Lagdon et al [30] examined the effect of CM on mental health outcomes among 640 university students and found that university students who reported CM had increased odds of depression, anxiety, and PTSD. An et al [33] conducted a network analysis of 476 college students with childhood abuse experiences and revealed that students who were abused in childhood reported a higher rate of co-morbidity of PTSD and depression compared to previous studies. CM is also known to be correlated with eating disorders and suicidal ideation [34–36]. Monteleone et al [37] demonstrated positive associations between childhood emotional abuse and neglect, and eating disorder. A meta-analysis of risk factors for suicidal ideation reported CM as one of the major predictors of suicidal ideation [23]. One of the recent studies explored the pathway from childhood bullying victimization to young adult depressive and anxiety symptoms in college students and showed that childhood traditional victimization escalated the risk of depression and anxiety, while childhood cyber-victimization increased the risk of anxiety [11].

A few studies have examined the association between the patterns of ACEs and internalizing symptoms using LCA and consistently demonstrated that the high/multiple ACEs class is related to higher levels of psychological symptoms, such as depression, compared to the low ACEs class [21,38]. However, there is still a dearth of studies that used LCA to identify the typology of CM and its associations with various internalizing symptoms among young adults in Asia. As the CM cases have increased in recent years [39], more research in Asian context is needed to better understand the patterns of CM and the psychological consequences of CM, especially among various Asian populations.

## The present study

The present study used LCA to determine distinct latent classes of CM and examined the associations between the classes of CM and five internalizing symptoms (i.e., depressive symptoms, anxiety symptoms, PTSD, eating disorder, and suicidal ideation) among young adults. It is hypothesized that several distinct classes of CM would be identified and that university

students in any of the CM classes would exhibit significantly higher likelihoods of internalizing symptoms when compared to those with low CM.

## Method

### Participants and data procedures

To test the research hypotheses, the present study employed a cross-sectional survey design and recruited university students from a major university in Singapore using purposive sampling. Participants were eligible for the study if they were full-time students between 21 and 30 years old, pursuing an undergraduate or graduate degree, using information and communications technology in everyday life, and English-speaking. Participants received an email invitation to participate in an online survey examining childhood and current trauma experiences, as well as mental health conditions. Data were collected via the university's eSurvey platform. The survey participation was anonymous and voluntary. Given the nature of an online survey, participant's informed consent was obtained in the following manner: Once the participants accessed the survey link, it led them first to a consent form, which outlines important details about the study, such as the purpose of the study, confidentiality, target population, the expected duration, and the possible benefits and risks associated with participation. After reading the consent form, respondents decided to participate in the survey and clicked the button "I Agree", indicating their informed consent and thereby leading them to the first page of the online survey (A waiver of the documentation of informed consent – no documented consent). That is, no documented consent form was collected. Instead, the participants consented to participate in the study by completing the enclosed online survey. The recruitment period for this study was between 1 August 2019 and 31 December 2019. After data collection, any personally identifiable information was deleted and non-identifiable data were securely stored in an encrypted drive. Prior to initiating the online survey, the study procedures were reviewed and approved by a university Institutional Review Board in accordance with the Declaration of Helsinki (NUS-IRB reference number: LS-18-201).

A final sample of 1,042 students aged 21-30 ($M_{age} = 23.8$, $SD = 1.81$) was included in this analysis. The majority were women (73%) and Chinese (92%). Nearly 43% of the participants' mothers obtained post-secondary education, followed by bachelor's degree (28%), less than post-secondary (18%), and post graduate diploma or master's degree (11%). The average monthly household income indicator was 4.73 ($SD = 1.62$; range from 1 = less than \$1000 to 7 = \$8001 and above; convert to average monthly income \$4,596).

### Measures

**Independent variable: Childhood maltreatment.** CM consists of seven domains of trauma experiences and bullying victimization in childhood (i.e., three domains of childhood abuse, two domains of childhood neglect, childhood traditional victimization, and childhood cyber victimization). The Childhood Trauma Questionnaire [40] was administered to assess five types of maltreatment (i.e., physical abuse, sexual abuse, emotional abuse, physical neglect, and emotional neglect) in childhood. Each subscale includes five items scored on a 5-point scale (1 = *never true* to 5 = *very often true*). Participants responded to each item in the context of "when you were growing up" (e.g., "I was punished with a belt, a board, a cord, or some other hard object"). Internal consistencies of the CTQ for our sample ranged from 0.60 for physical neglect to 0.93 for sexual abuse. Childhood traditional victimization (CTV) and cyber victimization (CCV) were measured by a single item for each (i.e., During childhood, "I was bullied via traditional bullying, such as physical, verbal, or relational bullying" and "I was bullied via cyberbullying"). The response categories ranged from 1 = *not at all* to 5 = *very often*.

All responses for the seven domains of CM were dichotomized into participants reporting that they *never* or *rarely* experienced maltreatment in childhood (no experience) and participants reporting that they *sometimes*, *often*, or *very often* experienced it (experience).

**Outcome variables: Internalizing symptoms.** Depressive symptoms were measured by a modified version of the 14-item Center for Epidemiologic Studies Depression Scale (CES-D) [41] derived from the CES-D [42]. Participants were asked to report how frequently each statement applied to them (e.g., "I felt bothered by things that usually don't bother me") in the past month, with four response options ranging from 1 = *rarely or none of the time* to 4 = *most or all of the time*. A total scale of the CES-D was created by averaging the responses across the items, with higher mean scores indicating higher levels of depressive symptoms (α=.89).

Anxiety symptoms were measured by the Beck Anxiety Inventory (BAI) [43]. The BAI is a 21-item self-report measure, assessing the extent of anxiety symptoms. Participants were asked to report how much they have been bothered by each symptom (e.g., "numbness or tingling," and "feeling of losing control") in the past month. All items were rated on a 4-point scale ranging from 0 = *not at all* to 3 = *severely*. These items were averaged to yield a mean score, with higher mean scores reflecting higher levels of perceived symptoms of anxiety (α=.93).

Post-traumatic stress disorder (PTSD) symptoms were assessed by the PTSD Checklist (PCL) [44], which is a validated 17-item self-rating scale assessing the intensity of PTSD symptoms (e.g., "In the past month, how much have you repeated, disturbing memories, thoughts, or images of the stressful experience"). All items were rated on a 5-point scale ranging from 1 = *not at all* to 5 = *extremely*. A composite indicator of the PCL was formed by averaging the responses on all the items, with higher mean scores reflecting higher levels of PTSD (α=.94).

Eating disorder was measured by the Eating Disorder Examination-Questionnaire (EDE-Q) (version 6.0) [45], which is a 28-item self-report measure that assesses the frequency of episodes of eating disorder psychopathology. It consists of four subscales, namely dietary restraint, eating concern, shape concern, and weight concern (e.g., "In the past 4 weeks, how many of days have you tried to exclude from your diet any foods that you like to influence your shape or weight?"). Items were scored using a 7-point rating scale (0–6), with scores of 5 or higher indicative of clinical range. The EDE-Q 6.0 generates a global score that is the average of the four subscale scores (α=.87), where a higher global score reflects more symptoms of eating disorders. The EDE-Q is a commonly used tool for assessing eating-disordered behaviors, developed from the Eating Disorder Examination interview (EDE) [45]. According to the American Psychiatric Association [APA], the EDE is considered the preferred instrument for assessing and diagnosing eating disorders, as outlined in the fourth edition of the Diagnostic and Statistical Manual of Mental Disorders (DSM-IV) [46].

Suicidal ideation was measured by the Suicidal Ideation Attributes Scale (SIDAS) [47]. The SIDAS consists of five items, each targeting an attribute of suicidal thoughts (e.g., "In the past month, how often have you had thoughts about suicide"). All items were rated on a 10-point scale, where a higher score reflects more severe suicidal thoughts (α=.79).

## Control variables

The five demographic variables were included as controls to account for the influence of background factors on the relationship between the patterns of CTEs and internalizing symptoms: age (in years); gender (*men*, *women*); race (*non-Chinese*, *Chinese*); maternal education level (*less than post-secondary* to *PhD*); and monthly household income (1 = *less than $1000*,

2 = $1001–$2000, 3 = $2001–$3500, 4 = $3501–$5000, 5 = $5001–$6500, 6 = $6501–$8000, 7 = $8001 and above).

## Statistical analyses

Data analyses consisted of the following phases: In the first phase, missing values were treated as Missing at Random (MAR) and were managed using the maximum likelihood estimation with robust standard errors. After imputing missing data, in the second phase, descriptive statistics were performed to examine the frequencies of all variables (i.e., social demographic characteristics, CM, and internalizing symptoms) by gender and ethnicity. In the third phase, to identify different classes of CM, latent class analysis (LCA) of the seven CM indicators was conducted using the "poLCA" package in *R* [48] and *Mplus* [49]. The best-fitting model was determined by examining the following model fit statistics: Akaike information criteria (AIC), Bayesian information criterion (BIC), adjusted BIC, entropy, and/or the Lo-Mendell Rubin (LMR) adjusted Likelihood Ratio Test (LRT) test. With regards to the BIC as "elbow method", lower values indicated better model fit. The classes in the final LCA model were named according to the observed item endorsement probabilities within each class. The predicted LCA class was saved as a new variable in the full dataset and participants who reported low CM was assigned a class value of 0, as they were utilized as the reference group in the main analyses. The final phase involved a series of multiple linear regression models to examine the association between the patterns of CM and internalizing symptoms (i.e., depressive symptoms, anxiety symptoms, PTSD, eating disorder, and suicidal ideation) while accounting for the effects of demographics, maternal education level, and family income.

## Results

### Descriptive statistics and comparisons by gender and ethnicity

As shown in Table 1, among sociodemographic characteristics, there were significant gender differences in participants' age and mother's education level, as well as ethnic difference in mother's education level: Men were older ($M_{age}$ = 24.3, $SD$ = 1.93) than women (Mean age = 23.4, $SD$ = 1.80; $t$ = -6.75, $p$ < .001). The average levels of mother's education were significantly higher in women (M = 2.41, $SD$ = 1.02) than men (M = 2.37, $SD$ = 1.11; $t$ = 3.43, $p$ < .001) and higher in non-Chinese students (M = 2.86, $SD$ = 1.37) than Chinese students (M = 2.32, $SD$ = 1.17; $t$ = 3.57, $p$ < .001).

The most common type of traumatic events was emotional neglect (74.6%) followed by emotional abuse (61%), physical neglect (55.6%), traditional victimization (46.6%), physical abuse (36.8%), cyber victimization (14.9%), and sexual abuse (10.7%). There were gender differences in childhood physical abuse (CPA) and childhood emotional abuse (CEA): Men (47.3%) experienced more CPA than women (32.9%; $\chi^2$ = 18.51, $p$ < .001), while women (64%) reported more CEA than men (53%; $\chi^2$ = 10.38, $p$ < .01). There were also ethnic differences in childhood sexual abuse (CSA) and childhood emotional neglect (CEN): Non-Chinese students (18.4%) showed more CSA than Chinese students (10.1%; $\chi^2$ = 5.03, $p$ < .05), while Chinese students (75.7%) experienced more CEN than non-Chinese students (60.5%; $\chi^2$ = 8.52, $p$ < .01).

The average scores of participants' internalizing symptoms were 2.12 ($SD$ = 0.48) for depressive symptoms, 1.56 ($SD$ = 0.42) for anxiety symptoms, 1.63 ($SD$ = 0.60) for PTSD, 11.22 ($SD$ = 4.58) for eating disorder, and 9.77 ($SD$ = 6.47) for suicidal ideation. Scores for anxiety symptoms ($t$ = 2.03, $p$ < .05) and eating disorder ($t$ = 6.01, $p$ < .001) were significantly higher in women than men. Except for suicidal ideation, four internalizing symptoms were significantly higher in non-Chinese students than Chinese students ($t$ = 2.95, $p$ < .01 for depressive

**Table 1. Prevalence of Social Demographic Characteristics, Childhood Maltreatment, and Young Adult Internalizing Symptoms by Gender and Race.**

| | Total (n = 1042) | Men (n = 281) | Women (n = 761) | Gender difference | Non-Chinese (n = 76) | Chinese (n = 966) | Ethnic difference |
|---|---|---|---|---|---|---|---|
| | n (%) or Mean ± SD | n (%) or Mean ± SD | n (%) or Mean ± SD | $\chi^2$ test/ t-test | n (%) or Mean ± SD | n (%) or Mean ± SD | $\chi^2$ test/ t-test |
| Age | 23.80 ± 1.81 | 24.30 ± 1.93 | 23.43 ± 1.80 | $t = -6.75$*** | 23.82 ± 1.84 | 23.75 ± 1.80 | $t = 1.69$ |
| Gender (ref. man) | 761 (73.0) | – | – | – | 52 (68.4) | 709 (73.3) | $\chi^2 = 0.48$ |
| Race (ref. non-Chinese) | 966 (92.7) | 257 (91.5) | 709 (93.2) | $\chi^2 = 0.48$ | – | – | – |
| Mother education | 2.40 ± 1.05 | 2.37 ± 1.11 | 2.41 ± 1.02 | $t = 3.43$*** | 2.86 ± 1.37 | 2.32 ± 1.17 | $t = 3.57$*** |
| Family income | 4.73 ± 1.62 | 2.74 ± 1.81 | 2.72 ± 1.58 | $t = 0.15$ | 4.40 ± 1.94 | 4.80 ± 1.88 | $t = -1.88$ |
| CPA | 383 (36.8) | 133 (47.3) | 250 (32.9) | $\chi^2 = 18.51$*** | 32 (42.1) | 351 (36.3) | $\chi^2 = 1.01$ |
| CSA | 112 (10.7) | 30 (10.7) | 82 (10.8) | $\chi^2 = 0.00$ | 14 (18.4) | 98 (10.1) | $\chi^2 = 5.03$* |
| CEA | 636 (61.0) | 149 (53.0) | 487 (64.0) | $\chi^2 = 10.38$** | 50 (65.8) | 586 (60.7) | $\chi^2 = 0.78$ |
| CPN | 579 (55.6) | 156 (55.5) | 423 (55.6) | $\chi^2 = 0.00$ | 35 (46.1) | 544 (56.3) | $\chi^2 = 3.01$ |
| CEN | 777 (74.6) | 212 (75.4) | 565 (74.2) | $\chi^2 = 0.16$ | 46 (60.5) | 731 (75.7) | $\chi^2 = 8.52$** |
| CTV | 486 (46.6) | 139 (49.5) | 347 (45.6) | $\chi^2 = 1.23$ | 39 (51.3) | 447 (46.3) | $\chi^2 = 0.72$ |
| CCV | 155 (14.9) | 46 (16.4) | 109 (14.3) | $\chi^2 = 0.68$ | 11 (14.5) | 144 (14.9) | $\chi^2 = 0.01$ |
| Depressive symptoms | 2.12 ± 0.48 | 2.07 ± 0.51 | 2.13 ± 0.47 | $t = -1.75$ | 2.29 ± 0.54 | 2.10 ± 0.47 | $t = 2.95$** |
| Anxiety symptoms | 1.56 ± 0.42 | 1.51 ± 0.45 | 1.58 ± 0.41 | $t = -2.03$* | 1.69 ± 0.51 | 1.55 ± 0.41 | $t = 2.34$* |
| PTSD | 1.63 ± 0.60 | 1.61 ± 0.63 | 1.64 ± 0.59 | $t = 0.76$ | 1.94 ± 0.78 | 1.61 ± 0.58 | $t = 3.71$*** |
| Eating disorder | 11.22 ± 4.58 | 9.88 ± 4.32 | 11.72 ± 4.58 | $t = 6.01$*** | 13.25 ± 5.26 | 11.06 ± 4.49 | $t = 3.53$** |
| Suicidal ideation | 9.77 ± 6.47 | 9.36 ± 7.16 | 9.92 ± 6.20 | $t = 1.15$ | 10.30 ± 8.13 | 9.72 ± 6.33 | $t = 0.61$ |

*Note*. Untransformed values for descriptive statistics. Transformed values used for t-tests/ $\chi^2$ tests. CPA = childhood physical abuse; CSA = childhood sexual abuse; CEA = childhood emotional abuse; CPN = childhood physical neglect; CEN = childhood emotional neglect; CTV = childhood traditional victimization; CCV = childhood cyber victimization.

\* $p < .05$, ** $p < .01$, *** $p < .001$.

symptoms; $t = 2.34$, $p < .05$ for anxiety symptoms; $t = 3.71$, $p < .001$ for PTSD; $t = 3.53$, $p < .01$ for eating disorder).

## Latent class analysis of childhood maltreatment

Table 2 describes the model fit statistics. The seven indicators of CM were entered into LCA ranging from 2 to 5 classes. We selected the 4-class solution as the best-fitting model for the optimal number of typologies, based on lowest AIC and BIC values and a significant LMR adjusted LRT (65.18, $p < .001$). Although the AIC and BIC values of the 5-class model were similar as those of the 4-class model, this model indicated no significant LMR adjusted LRT and one of the five classes was quite small (i.e., less than 5% of the total sample). As a result, the 4-class solution was deemed as the best-fitting and the most parsimonious model [50].

Fig 1 displays the 4-class model of CM and item-response probabilities for the seven CM indicators of each latent class. The four classes were named: *Low CM* (Class 1; 24.6%), *high/ multiple CM* (Class 2; 21.2%), *moderate to high abuse/victimization* (Class 3; 22.6%), and *moderate to high neglect* (Class 4; 31.6%). The class frequencies of CM across the four latent classes are presented in Table 3. Students in the *low CM* class (Class 1) had minimal or low probabilities of endorsing any CM. The *high/multiple CM* class (Class 2) was characterized by the highest probabilities of endorsing 4 out of 7 CM (CPA, CEA, CSA, CPN) and high probabilities of endorsing CEN, CTV, and CCV. The *moderate to high abuse/victimization* class (Class 3) was distinguished by moderate to high probabilities of endorsing childhood abuse

**Table 2. Fit Statistics for the Latent Class Analysis for 2-5 Classes.**

| | Number of classes | | | |
|---|---|---|---|---|
| Criteria | 2 | 3 | 4 | 5 |
| AIC | 7952.12 | 7868.41 | 7818.06 | 7818.54 |
| BIC | 8026.36 | 7982.23 | 7971.47 | 8011.54 |
| Adjusted BIC | 7978.71 | 7909.18 | 7873.01 | 7887.67 |
| Entropy | 0.610 | 0.583 | 0.581 | 0.669 |
| LMR adjusted LRT | 473.82 ($p < .001$) | 97.96 ($p < .001$) | 65.18 ($p < .001$) | 15.25 ($p = .293$) |
| $G^2$ | 270.70 | 170.99 | 104.64 | 89.12 |
| $l$ | −3961.06 | −3911.20 | −3878.03 | −3870.27 |

*Note*. AIC, BIC, adjusted BIC, and LMR adjusted LRT support the 4-class model. The 4-class was chosen.

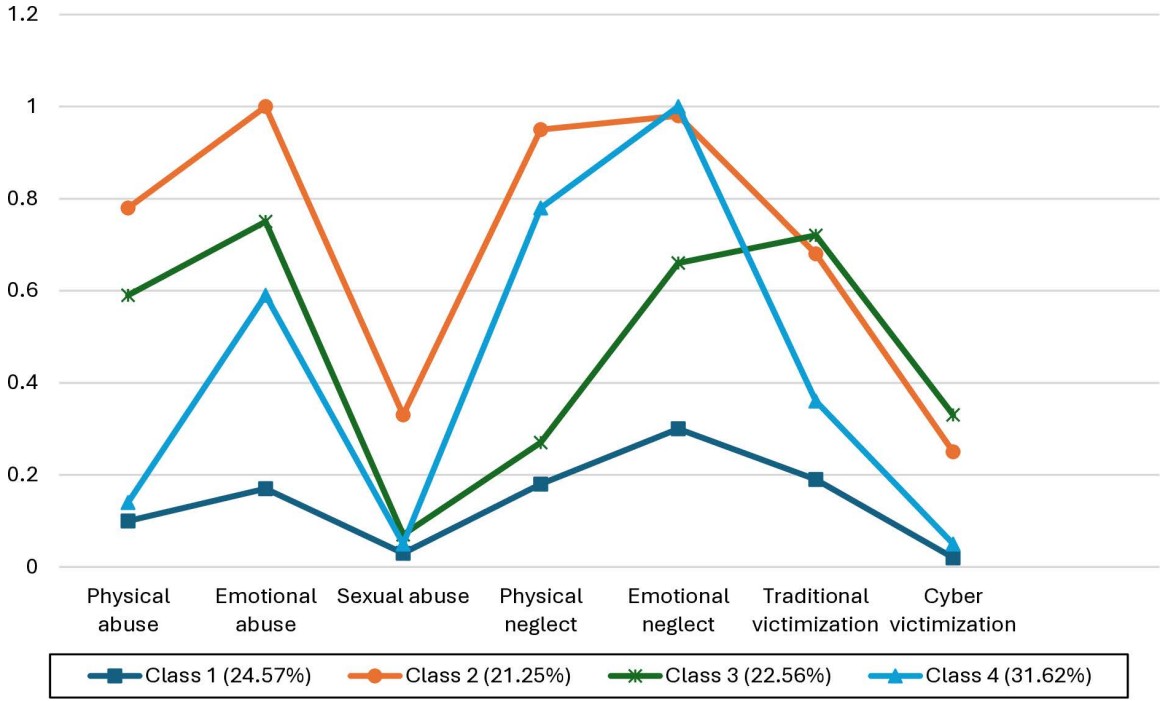

**Fig 1. Item-Response Probabilities for Seven Childhood Traumatic Experiences for the Four Latent Classes.**

(physical, emotional, sexual) and childhood victimization (traditional and cyberbullying), while displaying low probabilities of endorsing exposure to childhood neglect. Participants in the *moderate to high neglect* class (Class 4) had moderate or high probabilities of endorsing exposure to CPN and CEN, while displaying low probabilities of endorsing childhood abuse and victimization.

## Latent classes of childhood maltreatment and multiple mental health problems

The results shown in Table 4 were derived using a series of multiple linear regression models, which assessed the relationship between the latent classes of CM and five mental health

**Table 3. Class Frequencies of Childhood Maltreatment (4-Class Model).**

|        | Class 1 (24.57%) | Class 2 (21.25%) | Class 3 (22.56%) | Class 4 (31.62%) |
|--------|------------------|------------------|------------------|------------------|
| CPA    | .10              | .78              | .59              | .14              |
| CEA    | .17              | 1                | .75              | .59              |
| CSA    | .03              | .33              | .07              | .05              |
| CPN    | .18              | .95              | .27              | .78              |
| CEN    | .30              | .98              | .66              | 1                |
| CTV    | .19              | .68              | .72              | .36              |
| CCV    | .02              | .25              | .33              | .05              |

*Note.* CPA = childhood physical abuse; CSA = childhood sexual abuse; CEA = childhood emotional abuse; CPN = childhood physical neglect; CEN = childhood emotional neglect; CTV = childhood traditional victimization; CCV = childhood cyber victimization. Class 1 = low CM; Class 2 = high/multiple CM; Class 3 = moderate to high abuse/ victimization; Class 4 = moderate to high neglect.

**Table 4. Multiple Linear Regression Models Indicating the Relationship between the Classes of Childhood Maltreatment and Young Adult Internalizing Symptoms after Controlling for Social Demographic Characteristics.**

|                        | Depressive symptoms        | Anxiety symptoms           | PTSD symptoms              | Eating disorder           | Suicidal ideation        |
|------------------------|----------------------------|----------------------------|----------------------------|---------------------------|--------------------------|
|                        | *b* (95% CI)               | *b* (95% CI)               | *b* (95% CI)               | *b* (95% CI)              | *b* (95% CI)             |
| Age                    | −0.01 (−0.03, 0.01)        | −0.00 (−0.02, 0.01)        | −0.01 (−0.03, 0.01)        | −0.08 (−0.23, 0.08)       | −0.24* (−0.45, −0.02)    |
| Gender (ref. man)      | 0.09* (0.02, 0.16)         | 0.11*** (0.04, 0.17)       | 0.10* (0.01, 0.19)         | 2.20*** (1.51, 2.89)      | 0.35 (−0.61, 1.30)       |
| Race (ref. non-Chinese)| −0.13* (−0.24, −0.03)      | −0.12* (−0.21, −0.02)      | −0.26*** (−0.39, −0.12)    | −2.50*** (−3.54, −1.46)   | −0.32 (−1.75, 1.11)      |
| Mother education       | 0.03 (−0.01, 0.06)         | 0.03* (0.00, 0.05)         | 0.05** (0.02, 0.09)        | −0.01 (−0.30, 0.28)       | 0.20 (−0.21, 0.60)       |
| Family income          | −0.03** (−0.05, −0.01)     | −0.02** (−0.04, −0.01)     | −0.03** (−0.06, −0.01)     | −0.10 (−0.28, 0.07)       | −0.23 (−0.47, 0.01)      |
| CM (ref. Class 1)      |                            |                            |                            |                           |                          |
| Class 2                | 0.38*** (0.29, 0.46)       | 0.36** (0.29, 0.43)        | 0.44*** (0.34, 0.54)       | 1.95*** (1.16, 2.75)      | 6.22*** (5.12, 7.32)     |
| Class 3                | 0.25*** (0.16, 0.33)       | 0.16*** (0.09, 0.23)       | 0.22*** (0.12, 0.33)       | 1.35*** (0.56, 2.14)      | 2.82*** (1.72, 3.92)     |
| Class 4                | 0.18*** (0.10, 0.25)       | 0.10** (0.03, 0.16)        | 0.05 (−0.04, 0.14)         | 0.73* (0.01, 1.45)        | 2.44*** (1.44, 3.43)     |

*Note.* CTEs = childhood traumatic experiences; Class 1 = low CM; Class 2 = high/multiple CM; Class 3 = moderate to high abuse/victimization; Class 4 = moderate to high neglect.

\* *p* < .05, \*\* *p* < .01, \*\*\* *p* < .001.

outcomes. Regarding the covariates, older students were less likely to report suicidal ideation than younger counterparts. Women were more likely than men to report the symptoms of depression, anxiety, PTSD, and eating disorder, while Chinese students were less likely than non-Chinese students to experience the symptoms of depression, anxiety, PTSD, and eating disorder. Students whose mother's education level was high were more likely to report anxiety and PTSD symptoms compared to those whose mother's education level was low, while students in high-income families were less likely to experience the symptoms of depression, anxiety, and PTSD compared to those in low-income families.

After controlling for all covariates, students in all CM classes (Class 2: *high/multiple CM*, Class 3: *moderate to high abuse/victimization*, and Class 4: *moderate to high neglect*) were more likely to report symptoms of depression, anxiety, eating disorder and suicidal ideation, compared to those who had experienced low CM (reference class). Meanwhile, Class 2: *high/ multiple CM* ($b = 0.44$, 95% CI 0.34-0.54) and Class 3: *moderate to high abuse/victimization* ($b = 0.22$, 95% CI 0.12-0.33) were significantly associated with increased PTSD symptoms as compared to low CM, but Class 4: *moderate to high neglect* were not significantly associated with PTSD symptoms.

## Discussion

Increasing evidence suggests that CM has been identified as a crucial factor associated with a wide range of long-lasting internalizing problems. Notably, maltreatment experiences and conditions are likely to co-occur in different ways. This study used LCA to assess the heterogeneity underlying seven indicators of maltreatment experiences in childhood and their linkages with internalizing symptoms, such as depressive symptom, anxiety symptoms, PTSD, eating disorder, and suicidal ideation among university students in Singapore.

Results from the LCA model indicate four distinct classes of students who differed in accordance with the type and co-occurrence of CM: *Low CM*, *high/multiple CM*, *moderate to high abuse/victimization*, and *moderate to high neglect*. A large body of existing investigations used distinct indicators of CM and constantly found diverse classes of CM, often ranging from low CM to high/multiple CM [4,38,51]. Similarly, this study utilized seven different indicators of CM and found four latent classes of CM. Prior studies demonstrated that the majority of participants reported low CM and only a small number of participants had high/multiple CM [21,52]. However, the present study reports a remarkably higher prevalence of *high/multiple CM* (21.2%) compared to previous studies, such as 2.1% [19], 3.3% [21], and 7.2% [52]. This study also found higher rates of the *moderate to high neglect* class (31.6%) and *moderate to high abuse/victimization* class (22.6%). These findings are critical to understand different types of CM in the Singapore context and the fact that many Singaporean young adults experienced at least one type of maltreatment experiences as a child. In addition, these findings may provide a set of valid typologies of CM that can be used for other child maltreatment studies.

Classes of *high/multiple CM* and *moderate to high abuse/victimization* were strongly related to the five internalizing problems. Our findings showed that *high/multiple CM* are a strong predictor of later eating disorder and suicidal ideation, which are consistent with other studies that reported high levels and multiple types of CM being the key predictors of eating disorder and suicidal ideation [23,26,53]. Additionally, the study findings are consistent with previous research indicating that young adults with a history of CM are more likely to report internalizing symptoms than those with low CM exposure and that distinct classes of CM are related to the risk of different levels of internalizing symptoms [4,21,38,54,55]. Although the current study did not aim to identify the specific path from the history of CM to later mental health outcomes, a significant association between CM and mental health symptoms and outcomes further implies that childhood trauma, negative mental health symptoms, and psychiatric outcomes during adulthood can be explored in multiple directions with a set of mediator(s) or moderator(s). For example, CM was associated with poorer mental health functioning through negative posttraumatic cognitions among young adults [56]. There were also instances where physical maltreatment and binge eating was mediated by psychological distress (e.g., depression, anxiety) [57], and the relationship between emotional neglect or emotional abuse and suicide risk was mediated by depression [58]. As such, more studies should be conducted to test the various forms of directionality among childhood maltreatment, mental health, and the associated internalizing outcomes in order to develop the most

effective prevention and intervention programs that not only reduce traumatic experiences (e.g., abuse, neglect, victimization) for children, but also, boost the psychological well-being of young adult victims of child maltreatment.

Meanwhile, *moderate to high neglect* class was associated with the symptoms of depression, anxiety, eating disorder and suicidal ideation, but not significantly associated with PTSD symptoms. This aligns with the results of the meta-analysis, where pooled odds ratios for the association between neglect and depressive disorders, and neglect and anxiety disorders were significant, whereas its association with PTSD was not statistically significant [59]. One possible speculation is that neglect is different from other forms of child maltreatment, as it is closely tied to poverty. Since neglect usually involves the status when the caregivers are unable to meet the child's basic needs, it may not always result in the child experiencing trauma. In the similar vein, Dorahy and colleagues [60] postulated that physical neglect being the material deprivation and not necessarily emotional deprivation would have relatively less impact on children trauma-wise. Nevertheless, more evidence is required to clearly explain why physical and emotional neglect group may be related to depression and anxiety, but not with PTSD symptoms.

## Limitations and future research directions

Despite several robust strengths, the present study is not without limitations. The cross-sectional survey design poses limitations about the current analysis, as the temporal association between CM and internalizing symptoms in young adulthood cannot be determined. Future research using a longitudinal study design would be beneficial for examining the effects of the typologies of CM on young adult internalizing symptoms. The self-reported data may lead to under- or over-estimating experiences in the context of CM and internalizing symptoms. Specifically, a retrospective self-report of CM occurring before age 18 might have led to social desirability bias or distorted retrieval from memory [61]. To prevent this potential risk, future research would benefit from examining CM from multiple reporters. However, a large body of studies have demonstrated strong psychometric properties of CM indicators found no difference in the strength of relationship between CM and internalizing problems in adulthood on retrospective versus prospective reports of childhood adversity [21,38,62]. Further, the five internalizing symptoms (i.e., CES-D, BAI, PCL, EDE-Q 6.0, and SIDAS) in this study are self-report screening assessments, which are possibly of less consequence than other operationalizations of mental health disorders measured by clinicians or medical record diagnoses. In particular, this study did not explore the association of each subscale of eating disorder measure (EDE-Q 6.0) with CM. Therefore, future research would benefit from exploring how CM is related to different types of eating disorders. Due to limitations in obtaining samples, there are twice as many women than men in the study sample and more than 10 times as many Chinese than non-Chinese. The gender and race/ethnicity imbalance in the study sample could have impacted the results, and thus, the results cannot accurately represent the whole Singaporean sample. Future studies should develop a sampling strategy that could achieve proportionality in both gender and ethnicity for better interpretation. Furthermore, although the aim of the current study was not to identify the mediating factors associated with childhood maltreatment and internalizing behaviors, it would be worth exploring the mental health mediators based on the latent classes identified in this study, as there are only a handful number of studies that examined the mediating role of mental health in the association between childhood maltreatment and cognitive functioning [63,64].

## Implications for practice and policy

This study demonstrated that CM have detrimental long-term consequences on mental health, highlighting the importance of protecting children from CM and providing support to

victims in order to mitigate the effects of CM. The two most common CM among the participants were CEN (74.6%) followed by CEA (61%). Emotional maltreatment may be easier to commit compared to physical maltreatment, as the consequences of the former are non-tangible. Injuring children or neglecting their physical health are indicative of poor parenting, but the line is often blurred between disciplining children and emotionally abusing them. For instance, caregivers may put their children down as a way of "encouraging" them to do better. Especially, in the nature of Asian authoritarian parenting, many Asian parents have a lower awareness of emotional abuse and neglect [65,66]. Hence, aside from sharing alternative parenting strategies to physical punishment, parenting workshops must also cover how parents may communicate with children in a firm but respectful manner. Attendance to such workshops should be made mandatory for all caregivers of children and youth.

Further, men were more likely to experience CPA (47.3%) compared to women (32.9%). This corresponds with previous findings that boys tend to experience more harsh physical punishment than girls [34,65] because boys tend to exhibit more aggressive behaviors, and as a result, caregivers use harsher parenting methods on them [66]. However, caregivers may instead use hurtful remarks or psychological manipulation to "punish" girls, and this could explain why the proportion of girls who reported CEA (64%) was higher than boys (32.9%) in this study. Another possible explanation for the lower report of CEA by men could be the masculine gender script in the society that leads men to set a different threshold for emotional abuse and "hurtful or insulting things" as compared to women, as the former would believe they are considered "weak" if they took harsh words to heart. The risk of men downplaying the harm they have experienced engenders the need to relieve the social pressure to "be a man" and for national mental health campaigns to promote the message that men, too, can have emotional needs. More research is needed to determine whether women are indeed more susceptible to CEA than men. In the meantime, school curriculum for children of all ages should cover the difference between appropriate disciplining and emotional abuse so that children know when to voice out and protect themselves from further incidents.

Nearly three-quarters of the young adults in this study had suffered emotional neglect as children, highlighting the importance of identifying and tackling the reasons that caregivers neglect their children. Neglect may occur when caregivers are preoccupied with other matters such as long hours of work, caregiving to other members in the family, or personal mental health issues [28,67]. To reduce maladaptive caregiving behavior, caregivers' needs should be addressed and supported.

Along with preventive measures, such as early detection by school staff and education for children as well as caregivers, strategies to support youth facing CM and reduce the dire outcomes on their mental health should be cultivated. In doing so, researchers may focus on understanding potential protective factors for victims of CM, particularly those that can be introduced earlier in a child's development. For instance, Glickman et al [68] found that although CEN is a risk factor for later depression, strong peer social support at age 15 may reduce the risk of depressive symptoms by the time children reach late adolescence, suggesting that peer support could be one of the key factors to alleviate negative impacts of CEN. To increase the number of young adults who seek help for their internalizing symptoms, regular completion of anonymous, self-administered mental health assessments could be made mandatory for tertiary students on their school portals. A student who is at risk of or suffering from mental health conditions may then be redirected to helplines and counselling options.

Following the early detection, the current study also proposes the need for early intervention. According to Angelakis et al [69], many studies revealed that adults who have CM were more likely to have attempted or thought about suicide and that the individual's age was also a significant moderator in the relationship between child maltreatment and suicidality. As it

relates to eating disorder, bulimia nervosa and binge eating disorder were significantly associated with childhood abuse, such as sexual, physical, and emotional [70]. In this regard, mental health professionals should first assess the history of CM and provide early intervention programs to reduce the risk of suicide and eating disorder. Therapeutic support should also target non-clinical population in the community, as they are less exposed to mental health services than clinical population, but may still have untreated trauma in them [70].

## Conclusion

Our findings provide a profound understanding of the association between the four different patterns of CM and internalizing problems. In this study, the findings indicate that the classes of *high/multiple CM* and *moderate to high abuse/victimization* were both strongly related to all five internalizing problems, while the *moderate to high neglect* class was significantly associated with four internalizing symptoms, but not PTSD. Compared to the classes of *moderate to high abuse/victimization* and *moderate to high neglect*, the *high/multiple CM* class showed a higher level of young adult internalizing symptoms. However, most psychological intervention programs, such as the commonly employed trauma-focused cognitive behavioral therapy tend to be used similarly for victims of different patterns of CM. The mechanism behind why each victim suffers from each problem as well as the presentation of such symptoms would be different. Hence, any intervention would need to be targeted at the specific maltreatment experience and their associated internalizing symptoms to better understand the unique features of CM and promote the psychological well-being of young victims of CM. Along with these efforts, more thorough research would be necessary to examine the long-term effects of the different typologies of CM on behavioral and psychological problems.

## Supporting information

**S1 File. Supporting Infomation-BBHS Survey Data.**
(XLSX)

## Author contributions

**Conceptualization:** Jungup Lee.

**Data curation:** Jungup Lee.

**Formal analysis:** Jungup Lee.

**Funding acquisition:** Jungup Lee.

**Methodology:** Jungup Lee.

**Supervision:** Jungup Lee.

**Validation:** Jungup Lee, Jinyung Kim.

**Visualization:** Jungup Lee.

**Writing – original draft:** Jungup Lee, Yogeswari Munisamy, Tan Ai, Sungwon Yoon, Jinyung Kim, Alicia Pon.

**Writing – review & editing:** Jungup Lee, Yogeswari Munisamy, Tan Ai, Sungwon Yoon, Jinyung Kim, Alicia Pon.

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
