## [Decision Letter · Decision Letter 0]

16 Sep 2024

PONE-D-24-24662Typologies of Childhood Traumatic Experiences and Associations with Internalizing Symptoms among Young Adults in Singapore: A Latent Class AnalysisPLOS ONE

Dear Dr. Lee,

Thank you for submitting your manuscript to PLOS ONE. After careful consideration, we feel that it has merit but does not fully meet PLOS ONE’s publication criteria as it currently stands. Therefore, we invite you to submit a revised version of the manuscript that addresses the points raised during the review process.

We look forward to receiving your revised manuscript.

Kind regards,

Vanessa Carels

Staff Editor

PLOS ONE

“This work was supported by Start-Up Grant from the National University of Singapore [Grant No. R134-000-098-133].”

3. In this instance it seems there may be acceptable restrictions in place that prevent the public sharing of your minimal data. However, in line with our goal of ensuring long-term data availability to all interested researchers, PLOS’ Data Policy states that authors cannot be the sole named individuals responsible for ensuring data access (http://journals.plos.org/plosone/s/data-availability#loc-acceptable-data-sharing-methods).

6. Please include a separate caption for each figure in your manuscript.

Comments from PLOS Editorial Office: We note that one or more reviewers has recommended that you cite specific previously published works. As always, we recommend that you please review and evaluate the requested works to determine whether they are relevant and should be cited. It is not a requirement to cite these works. We appreciate your attention to this request.

Reviewers' comments:

Reviewer's Responses to Questions

**Comments to the Author**

1. Is the manuscript technically sound, and do the data support the conclusions?

Reviewer #1: Yes

Reviewer #2: Yes

Reviewer #3: Yes

2. Has the statistical analysis been performed appropriately and rigorously? 

Reviewer #1: Yes

Reviewer #2: I Don't Know

Reviewer #3: Yes

3. Have the authors made all data underlying the findings in their manuscript fully available?

Reviewer #1: Yes

Reviewer #2: Yes

Reviewer #3: No

4. Is the manuscript presented in an intelligible fashion and written in standard English?

Reviewer #1: Yes

Reviewer #2: Yes

Reviewer #3: Yes

5. Review Comments to the Author

Reviewer #1: This study used Latent Class Analysis to identify typologies of childhood traumatic experiences and the associations of CTEs with five internalizing symptoms. The research has a certain academic value and can arouse readers’ interest, but there are still some problems that the author needs to pay attention to.

1. There are few references in recent five years, so it is recommended to check and update the literature to ensure the comprehensiveness and timeliness of the literature review.

2. In the introduction, the author mentions the use of Asian samples as one of the uniqueness and value of the study, but in the paper, there is less explanation of the comparison and possible mechanism of the similarities and differences between Eastern and Western subjects.

3. With more than twice as many women as men in the study sample and more than 10 times as many Chinese as non-Chinese, there is an imbalance in the grouping that may have had some influence on the results.

4. There are cases where writing is not rigorous, for example, page 4, latent class analysis (LCA) mixed with LCA; Page 8, The average monthly household income was 4.73 (SD=1.62; range 1-7). Lack of units of average monthly income.

Reviewer #2: Summary

This manuscript utilized a latent class analysis to explore childhood traumatic experiences and outcomes on several mental health domains. The authors specifically examined the four classes that emerged and their relationship to depression, anxiety, eating disorders, PTSD, and suicidality. This manuscript offers an important perspective of these experiences in an Asian population, which is necessary, given the saturation of research in Western samples. However, the manuscript can be improved by addressing the following comments. Most importantly, more nuanced in interpretation of findings is necessary, as well as specific attention to the method of analysis and interpretation of the classes conflated with different types of CTEs.

Introduction

- The intro discusses that CTEs are related to increased eating disorder rates, but I’m wondering if there will be an exploration of which EDs, given there is a diversity of them (e.g., anorexia to binge eating). This is also relevant for the methods section when the EDQ is discussed. I’m wondering if that has been used to assess all Eds, or those more related to body image and restriction, like anorexia or bulimia. However, EDs like Avoidant Restriction Food Intake Disorder (ARFID) and binge eating may not be as relevant. All this to say it may be important to classify or operationalize EDs more specifically throughout the manuscript.

Methods

- Women/men would be more appropriate if gender is being explored, whereas male/female would fit best if sex is the demographic variable of interest

Results

- It states “Insert Table 1, about here” on p. 12 of the manuscript. I’m unsure if this is an error or a formatting reference to Tables in the supplemental documents. Similar for tables and figures later on

- The presentation of covariates relationship to the outcome variables would make sense to present prior to the linear regression results that contain the demographic variables as covariates

- Is it assumed that the 4 classes provided by the LCA are mutually exclusive? It can be that participants can fall into the categories of high/multiple CTEs and also abuse/victimization, for example. How does that influence data classification and interpretation?

- Is there a way to classify, or provide more information people who may fall in more than one typology as described by the LCA?

Discussion

- It is a bit confusing to refer to the four latent classes as different typologies of CTE, given that two refer to frequency (high vs low) and refer to type of trauma experienced (neglect or abuse). I’m wondering if there is a different way to classify those, or if more explanation is warranted as to how those are characterized into typologies of CTE. Typologies of CTE may render readers to think more about CTQ subscales (neglect, abuse, physical, sexual, etc). It might make more sense to discuss charcaterestics of CTEs.

- As authors discuss, associations are not a direct path. Thus, implications about sequential relationship between the variables may not be fully appropriate given the method of analysis and what is missing to attribute causality or even directionality. For example, it could be that individuals with existing mental health challenges elicit greater neglect or abuse from caregivers due to the stress of caring for them, or it could also be the other relationship. That nuanced, bidirectional relationship should be explored given the method of analysis rather than assuming the direction can be exacted with associations alone.

- The discussion of eating disorders can be made more nuanced and details given the diverse presentation of EDs (see my earlier comment in introduction)

- The discussion of mediators are helpful, but again limited given the study design.

- the discussion briefly mentions how these results differ from western studies. This is important, but also requires more explanation as to why that is, or how it’s believed that these may be different.

- I’m unsure that physical, material deprivation is not related to negative outcomes at all. I would specify that this is related to PTSD only as negative findings, but not with depression and anxiety. I think more information regarding why authors can understand the link between neglect and depression/anxiety would be helpful.

Reviewer #3: Thank you for providing me with this opportunity to review such a wonderful manuscript.

Title: The authors are advised to replace childhood traumatic experiences with Childhood maltreatment or Adverse childhood events/experiences.

Abstract: I miss to see the measurement/assessment tools used in the study.

Review of Literature

The literature review provides a broad overview of the global and regional significance of of identifying typologies of childhood traumatic experiences (CTEs) and their associations with CTEs with five internalizing symptoms in 1,042 university students in Singapore. It contextualizes the issue within the broader public health crisis and highlights the specific vulnerability of survivors of CTEs.

Although the review covers many relevant studies, it could include more recent works in the field of childhood maltreatment and trauma (eg. https://doi.org/10.1186/s13034-021-00373-7,
https://doi.org/10.1080/20008198.2021.2007730)

Identification of the gap

The manuscript correctly identifies the gap in the literature regarding childhood advance experiences. The focus on this specific population and the attempt to explore “Childhood Traumatic experiences”and their association with both externalized and internalized symptoms using latent classic analysis is commendable. The manuscript should however, clearly articulate why filling this gap is crucial for both academic knowledge and practical interventions.

Methods

The manuscript misses structural methodological practices (e.g., data entry and security, ethical considerations, design). The authors could consider including some of these subheadings.

The study uses well-established and standardized instruments like the“Center for Epidemiological Studies Depression Scale” Beck Anxiety Inventory, which adds to the methodological rigor. In relation to each measure, the authors should discuss reliability in this specific context.

It is shown that PTSD was assessed by a checklist. The authors could mention a specific name for this tool. What do the authors mean by PCL in the same paragraph?

Childhood traumatic experiences could be replaced by either maltreatment or Advance Childhood experiences. Maltreatment or ACES might not necessarily be traumatic.

Since the authors have opted to have a sub heading on the control variables, it would be beneficial `to have subheading on both Predictor and outcome variables as well.

Results: The type of the regression model utilized in the analysis should be clearly stated in the analysis plan and table headings.

I appreciate the fact that the current table 1 presents the mean scores for the main study variables. However, the manuscript misses information on the social demographic characteristics.

resenting results on social demographic characteristics and intercorrelations between the continuous variables would benefit the manuscript.

Discussion

The discussion is well written

Overall Assessment

This manuscript addresses an important subject of advance childhood experiences among the university students in Singapore. The study’s use of established standardized tools and appropriate statistical analyses adds rigor to the findings.

6. PLOS authors have the option to publish the peer review history of their article (what does this mean? ). If published, this will include your full peer review and any attached files.

**Do you want your identity to be public for this peer review?** For information about this choice, including consent withdrawal, please see our Privacy Policy .

Reviewer #1: No

Reviewer #2: No

Reviewer #3: **Yes: ** HERBERT AINAMANI

---

## [Author Response · Author response to Decision Letter 1]

6 Feb 2025

We have also uploaded the response letter as an attachment.

Dear Editor,

Thank you for this opportunity to revise and resubmit the manuscript entitled “Typologies of Childhood Maltreatment and Associations with Internalizing Symptoms among Young Adults in Singapore: A Latent Class Analysis” (PONE-D-24-24662) to PLOS ONE. We appreciate the reviewer’s comments which have guided us to improve the quality of the paper. In this letter, the authors are providing point-by-point responses to all the reviewer’s comments.

This revised manuscript contains original material and has not been submitted for review elsewhere. If I can provide any information that would be of use to editorial staff, please do not hesitate to contact me.

Thank you in advance for your time and effort in the review of this manuscript, and I appreciate your consideration. I look forward to hearing from you soon.

Response to the editorial comments

[Author Response]: We carefully followed PLOS ONE’s style. Thank you!

“This work was supported by Start-Up Grant from the National University of Singapore [Grant No. R134-000-098-133].”

[Author Response]: We have amended the Role of Funder statement in our cover letter by including the statement, “The funders had no role in study design, data collection and analysis, decision to publish, or preparation of the manuscript."

3. In this instance it seems there may be acceptable restrictions in place that prevent the public sharing of your minimal data. However, in line with our goal of ensuring long-term data availability to all interested researchers, PLOS’ Data Policy states that authors cannot be the sole named individuals responsible for ensuring data access (http://journals.plos.org/plosone/s/data-availability#loc-acceptable-data-sharing-methods).

[Author Response]: There is no non-author institutional point of contact to access the data. It is up to the discretion of the first author to give permission to the data used in this study. As a result, we added the following statement for those who may be interested in obtaining this data at the end of the manuscript:

“Availability of data and materials

The data that support the findings of this study are available by the first author, Jungup Lee, but restrictions apply to the availability of these data due to the issue of confidentiality. The data are, however, available from the first author upon reasonable request.”

[Author Response]: Amended the title identically.

[Author Response]: The ethics statement is already included in the Methods section. However, the authors added the IRB number and how the consent was obtained from the individuals (pp.7-8). The ethics statement which was initially located at the bottom of the manuscript is now deleted and it is only being discussed in the Methods section.

“To test the research hypotheses, the present study employed a cross-sectional survey design and recruited university students from a major university in Singapore using purposive sampling. Participants were eligible for the study if they were full-time students, between 21 and 30 years old, pursuing an undergraduate or graduate degree, using information and communications technology in everyday life, and English-speaking. Participants received an email invitation to participate in an online survey examining childhood and current trauma experiences, as well as mental health conditions. Data were collected via the university’s eSurvey platform. The survey participation was anonymous and voluntary. Given the online survey, participants’ informed consent was obtained in the following manner. Once the participants accessed the survey link, it led them first to a consent form, which outlines important details about the study, such as the purpose of the study, confidentiality, target population, the expected duration, and the possible benefits and risks associated with participation. If, after reading the consent form in full, respondents decided to participate in the survey, they clicked the “I Agree” button, indicating their informed consent and thereby leading them to the first page of the online survey (A waiver of the documentation of informed consent – no documented consent). That is, no documented consent form was collected. Instead, the participants consented to participate in the study by completing the enclosed online survey. The recruitment period for this study was between 1 August 2019 and 31 December 2019. After data collection, any personally identifiable information was deleted, and non-identifiable data were securely stored in an encrypted drive. Prior to initiating the online survey, study procedures were reviewed and approved by a university Institutional Review Board in accordance with the Declaration of Helsinki (NUS-IRB reference number: LS-18-201).”

6. Please include a separate caption for each figure in your manuscript.

[Author Response]: There is only one figure used in this manuscript and the authors added the caption for this figure as follows:

“<<Insert Figure-1: Item-Response Probabilities for Seven Childhood Traumatic Experiences for the Four Latent Classes, about here>>”

Comments from PLOS Editorial Office: We note that one or more reviewers has recommended that you cite specific previously published works. As always, we recommend that you please review and evaluate the requested works to determine whether they are relevant and should be cited. It is not a requirement to cite these works. We appreciate your attention to this request.

[Author Response]: We make sure to review the works before citing them in our manuscript. Thank you.

Response to the reviewer’s comments

Reviewer 1:

1. There are few references in recent five years, so it is recommended to check and update the literature to ensure the comprehensiveness and timeliness of the literature review.

[Author Response]: Thanks for your advice. We have updated with several new references as per your comments. Here are some of the recent references we added to the Introduction:

Hecker T, Boettcher VS, Landolt MA, Hermenau K. Child neglect and its relation to emotional and behavioral problems: A cross-sectional study of primary school-aged children in Tanzania. Development and Psychopathology. 2018 Mar;31(1):325-339.

Ainamani HE, Weierstall-Pust R, Bahati R, Otwine A, Tumwesigire S, Rukundo G. Post-traumatic stress disorder, depression and the associated factors among children and adolescents with a history of maltreatment in Uganda. European Journal of Psychotraumatology. 2022 Jan;13(1):1-11.

Henry LM, Gracey K, Shaffer A, Ebert J, Kuhn T, Watson KH, et al. Comparison of three models of adverse childhood experiences: Associations with child and adolescent internalizing and externalizing symptoms. Journal of Abnormal Psychology. 2021 Jan;130(1):9-25.

Bevilacqua L, Kelly Y, Heilmann A, Priest N, Lacey RE. Adverse childhood experiences and trajectories of internalizing, externalizing, and prosocial behaviors from childhood to adolescence. Child Abuse Neglect. 2021 Feb;112:104890.

Leban L. The effects of adverse childhood experiences and gender on developmental trajectories of internalizing and externalizing outcomes. Crime & Delinquency. 2021 Jan;67(5):1-24.

Grigsby TJ, Rogers CJ, Albers LD, Benjamin SM, Lust K, Eisenberg ME, et al. Adverse childhood experiences and health indicators in a young adult, college student sample: Differences by gender. International Journal of Behavioral Medicine. 2020 Jul;27:660-667.

2. In the introduction, the author mentions the use of Asian samples as one of the uniqueness and value of the study, but in the paper, there is less explanation of the comparison and possible mechanism of the similarities and differences between Eastern and Western subjects.

[Author Response]: Thank you for the comment. The authors would like to clarify that the inclusion of Asian samples is not the unique feature of this study. Since the current statement is misleading, we made the following adjustments (pp.3-4): “For example, only a few studies determined the relationship between the typologies of adverse childhood experiences (ACEs) and internalizing problems/disorders (see Lew & Xian4 for U.S. children).”

3. With more than twice as many women as men in the study sample and more than 10 times as many Chinese as non-Chinese, there is an imbalance in the grouping that may have had some influence on the results.

[Author Response]: Thank you for the feedback. The authors are fully aware that the study sample did not achieve gender and ethnicity proportionality, which may have impacted the results and the interpretation of the findings. As a result, the authors included this as a limitation of the study (p.22):

“Due to limitations in obtaining samples, there are twice as many females as males in the study sample and more than 10 times as many Chinese as non-Chinese. There is an imbalance in the groupings which could impact the results and thus, the results cannot accurately represent the Singaporean sample. Future studies should develop a sampling strategy that could achieve proportionality in both gender and ethnicity.”

4. There are cases where writing is not rigorous, for example, page 4, latent class analysis (LCA) mixed with LCA; Page 8, The average monthly household income was 4.73 (SD=1.62; range 1-7). Lack of units of average monthly income.

[Author Response]: Thanks for your comments. We have revised the cases to have their full word and acronym rigorously (e.g., latent class analysis and LCA, adverse childhood experiences and ACEs). In addition, we have updated the average monthly household income.

Reviewer 2:

Introduction

1. The intro discusses that CTEs are related to increased eating disorder rates, but I’m wondering if there will be an exploration of which EDs, given there is a diversity of them (e.g., anorexia to binge eating). This is also relevant for the methods section when the EDQ is discussed. I’m wondering if that has been used to assess all Eds, or those more related to body image and restriction, like anorexia or bulimia. However, EDs like Avoidant Restriction Food Intake Disorder (ARFID) and binge eating may not be as relevant. All this to say it may be important to classify or operationalize EDs more specifically throughout the manuscript.

[Author Response]: Thank you for your insightful comment. The current study did not use a clinical scale to measure EDs, as the participants are not from a clinical population. ED was assessed by the 28-item EDE-Q (version 6.0) that consists of 4 subscales: dietary restraint, eating concern, shape concern, and weight concern), and we used a global score that is the average of the 4 subscale scores to measure university students’ eating disorder symptoms. The measurement for eating disorders is a questionnaire that has high internal consistency and test-retest reliability, and it is used in a few other published studies that are reflected in the Introduction and Methods (under Measures) sections. It is also considered the preferred instrument for assessing and diagnosing eating disorders, as outlined in the fourth edition of the Diagnostic and Statistical Manual of Mental Disorders (DSM-IV).

In this study, the authors did not explore the relationship between each subscale of the questionnaire and CM, and as such, will not be discussing this in the Introduction and Discussion sections. However, we note that different types of EDs may have different relationships with CM, and this would be an interesting area of exploration for future studies. We have included this under the Limitations section (see pp. 21-22).

“In particular, this study did not explore the association of each subscale of eating disorder measure (EDE-Q 6.0) with CM. Therefore, future research would benefit to explore how CM is related to different types of eating disorders.”

Methods

2. Women/men would be more appropriate if gender is being explored, whereas male/female would fit best if sex is the demographic variable of interest

[Author Response]: Thank you for the advice. As gender was being explored in this study, we have amended to women/men in the manuscript accordingly.

Results

3. It states “Insert Table 1, about here” on p. 12 of the manuscript. I’m unsure if this is an error or a formatting reference to Tables in the supplemental documents. Similar for tables and figures later on

[Author Response]: Thank you for the comment. Now, all the Tables are located within the text instead of being placed at the bottom of the manuscript.

4. The presentation of covariates relationship to the outcome variables would make sense to present prior to the linear regression results that contain the demographic variables as covariates.

[Author Response]: Thank you for the comment. We first presented the relationship between covariates and the outcome variables and then presented the main linear regression results after controlling for the demographic variables.

5. Is it assumed that the 4 classes provided by the LCA are mutually exclusive? It can be that participants can fall into the categories of high/multiple CTEs and also abuse/victimization, for example. How does that influence data classification and interpretation?

[Author Response]: Thanks for your comment. In the current study, LCA assigned individual participants to the most probable latent class based on their response patterns. It means that the identified classes are mutually exclusive within the model. But this does not imply that individual participants are not able to exhibit characteristics of multiple classes. Instead, LCA categorizes them according to predominant response patterns t

---

## [Decision Letter · Decision Letter 1]

26 Feb 2025

Typologies of Childhood Maltreatment and Associations with Internalizing Symptoms among University Students in Singapore: A Latent Class Analysis

PONE-D-24-24662R1

Dear Dr. Lee,

We’re pleased to inform you that your manuscript has been judged scientifically suitable for publication and will be formally accepted for publication once it meets all outstanding technical requirements.

Kind regards,

Serkan Yılmaz

Academic Editor

PLOS ONE

Additional Editor Comments (optional):

Reviewers' comments:

Reviewer's Responses to Questions

**Comments to the Author**

1. If the authors have adequately addressed your comments raised in a previous round of review and you feel that this manuscript is now acceptable for publication, you may indicate that here to bypass the “Comments to the Author” section, enter your conflict of interest statement in the “Confidential to Editor” section, and submit your "Accept" recommendation.

Reviewer #2: All comments have been addressed

Reviewer #3: All comments have been addressed

2. Is the manuscript technically sound, and do the data support the conclusions?

Reviewer #2: Yes

Reviewer #3: Yes

3. Has the statistical analysis been performed appropriately and rigorously? 

Reviewer #2: Yes

Reviewer #3: Yes

4. Have the authors made all data underlying the findings in their manuscript fully available?

Reviewer #2: Yes

Reviewer #3: No

5. Is the manuscript presented in an intelligible fashion and written in standard English?

Reviewer #2: Yes

Reviewer #3: Yes

6. Review Comments to the Author

Reviewer #2: The authors addressed all concerns from the original submission. It has improved the manuscript and streamlined concerns.

Reviewer #3: The authors have responded to all my concerns satisfactorily and I have no reservation about the publication of

this manuscript in your journal

7. PLOS authors have the option to publish the peer review history of their article (what does this mean? ). If published, this will include your full peer review and any attached files.

**Do you want your identity to be public for this peer review?** For information about this choice, including consent withdrawal, please see our Privacy Policy .

Reviewer #2: No

Reviewer #3: **Yes: ** HERBERT E. AINAMANI

---

## [Editor Report · Acceptance letter]

PONE-D-24-24662R1

PLOS ONE

Dear Dr. Lee,

I'm pleased to inform you that your manuscript has been deemed suitable for publication in PLOS ONE. Congratulations! Your manuscript is now being handed over to our production team.

Kind regards,

on behalf of

Dr. Serkan Yılmaz

Academic Editor

PLOS ONE